# Novel Transcriptome Study and Detection of Metabolic Variations in UV-B-Treated Date Palm (*Phoenix dactylifera* cv. Khalas)

**DOI:** 10.3390/ijms22052564

**Published:** 2021-03-04

**Authors:** Mohamed Maher, Hasan Ahmad, Elsayed Nishawy, Yufei Li, Jie Luo

**Affiliations:** 1National Key Laboratory of Crop Genetic Improvement and National Center of Plant Gene Research (Wuhan), Huazhong Agricultural University, Wuhan 430070, China; mohamed13@webmail.hzau.edu.cn (M.M.); hasanngb@webmail.hzau.edu.cn (H.A.); elnishawy@mail.hzau.edu.cn (E.N.); yf.li@webmail.hzau.edu.cn (Y.L.); 2Department of Biochemistry, College of Agriculture, Zagazig University, Zagazig 44511, Egypt; 3National Gene Bank, Agricultural Research Center (ARC), Giza 12619, Egypt; 4Desert Research Center, Genetics Resource Department, Egyptian Deserts Gene Bank, Cairo 11735, Egypt; 5Institute of Tropical Agriculture and Forestry of Hainan University, Haikou 570288, China

**Keywords:** date palm, *Phoenix dactylifera*, transcriptome analysis, amino acids and secondary metabolites

## Abstract

Date palm (*Phoenix dactylifera*) is one of the most widespread fruit crop species and can tolerate drastic environmental conditions that may not be suitable for other fruit species. Excess UV-B stress is one of the greatest concerns for date palm trees and can cause genotoxic effects. Date palm responds to UV-B irradiation through increased DEG expression levels and elaborates upon regulatory metabolic mechanisms that assist the plants in adjusting to this exertion. Sixty-day-old Khalas date palm seedlings (first true-leaf stage) were treated with UV-B (wavelength, 253.7 nm; intensity, 75 μW cm^−2^ for 72 h (16 h of UV light and 8 h of darkness). Transcriptome analysis revealed 10,249 and 12,426 genes whose expressions were upregulated and downregulated, respectively, compared to the genes in the control. Furthermore, the differentially expressed genes included transcription factor-encoding genes and chloroplast- and photosystem-related genes. Liquid chromatography-tandem mass spectrometry (LC-MS/MS) was used to detect metabolite variations. Fifty metabolites, including amino acids and flavonoids, showed changes in levels after UV-B excess. Amino acid metabolism was changed by UV-B irradiation, and some amino acids interacted with precursors of different pathways that were used to synthesize secondary metabolites, i.e., flavonoids and phenylpropanoids. The metabolite content response to UV-B irradiation according to hierarchical clustering analysis showed changes in amino acids and flavonoids compared with those of the control. Amino acids might increase the function of scavengers of reactive oxygen species by synthesizing flavonoids that increase in response to UV-B treatment. This study enriches the annotated date palm unigene sequences and enhances the understanding of the mechanisms underlying UV-B stress through genetic manipulation. Moreover, this study provides a sequence resource for genetic, genomic and metabolic studies of date palm.

## 1. Introduction

Date palm (*Phoenix dactylifera*) is one of the most widespread fruit crop species in several arid and semiarid countries, i.e., those in North Africa, the Middle East and Central America [1]. The cultivar Khalas is one of the most common cultivars worldwide, and its fruit is among the best of the soft fruit types [2,3,4]. Date palm can tolerate drastic environmental conditions that may not be suitable for other fruit species [5]. In full sun or partial shade, date palm trees grow and are tolerant to drought. In any kind of soil, even relatively salty soils, they can thrive, but need good drainage. While the date palm is adaptable to various growing conditions, date fruit is produced only in dry and hot climates [6,7]. Date palm growth may be hindered by the violet and yellow ends of the electromagnetic spectrum but may be enhanced by rays at the other end of the spectrum.

Increasing amounts of UV-B radiation have reached the Earth’s surface and the surfaces of plants due to stratospheric ozone layer degradation [8]. UV-B radiation is a powerful environmental factor that affects many aspects of the life of a plant, including development, growth, and morphology. [9,10]. Numerous reports have shown that date palm trees, UV-A (320 nm to 400 nm) and UV-B (290 nm to 320 nm) are of the greatest concern. Excess UV-B rays can cause genotoxic effects, which are linked to the ability of UV-B to induce direct DNA damage in date palm [11]. UV-B damage involves the direct formation of thymine dimers or other dimers of pyrimidine and double-stranded DNA breaks. [11,12].

UV-B stress has caused plants to improve ways of modifying their metabolism and reconfiguring their internal networks of metabolism. Under UV radiation, plants produce high levels of many amino acids and some of their derivatives, such as Glu, Ala, Lys, Phe and γ-amino butyrate [13]. This altered activity, occurring after UV-B treatment at the primary-metabolite stage and leading to the synthesis of functional secondary metabolites that can defend against UV-B damage, has improved. [13,14]. UV-B is also considered a source of oxidative stress [15,16,17]. Additionally, UV-B leads to increased glutathione (GSH) and multiple alkaloid content, as well as increased activity of the superoxide dismutase (SOD) and peroxidase (POD) enzymes, which can be due to the generation of reactive oxygen species (ROS) [18].

The date palm response to UV radiation depends on secondary metabolite accumulation [6,19]. In addition, the amount of change occurring in the phenylpropanoid pathway determines the harmful effects of date palm caused by UV radiation [13,15]. Furthermore, flavonoids might function as antioxidants during photoprotection [20,21].

Despite the availability of a reference genome [22,23] along with a genetic map of date palm [24], there are limited expression analysis studies on date palm in response to UV-B stress [25,26,27]. Further physiological and molecular studies are, therefore, needed to explain the mechanisms underlying date palm adaptation to UV radiation.

Date palm has natural ecological survival ability, making the genome of this species valuable as a unique source of genes that may be involved in abiotic (UV radiation) stress. However, few studies have been conducted on the effects of UV stress on date palm. Therefore, the aim of this work was to identify UV stress-response genes to establish a useful database of information about the genome of date palm, as well as that of other crop plant species. The current research is the first study on date palm gene expression profiling in response to solar UV stress using high-throughput sequencing of RNA (RNA-seq).

## 2. Results

### 2.1. Sequence and De Novo Assembly of the Transcriptome of Khalas Date Palm under UV Stress Conditions

The RNA-seq high-throughput sequencing approach was used to sequence the date palm transcripts expressed in response to UV stress in four libraries: Khalas control 1 (CR1), Khalas control 2 (CR2), Khalas UV treatment 1 (UR1), and Khalas UV treatment 2 (UR2). All of them are biological replicates. These transcriptome sequencing libraries were prepared from the first leaf of 60-day-old *P. dactylifera* cv. Khalas plants treated with UV-B (wavelength, 253.7 nm; intensity, 75 μW cm^−2^) for 72 h. The four libraries were separately sequenced using the Illumina high-throughput second-generation sequencing platform. After removing the low-quality reads and all possible contaminants, 43,537,934, 44,128,934, 43,871,714 and 44,853,794 total reads and 41,339,168, 41,716,324, 41,324,006 and 42,319,388 mapped reads were obtained from CR1, CR2, CR3 and CR4, respectively. The proportion of nucleotides with a quality value >30 ranged from 93% to 94%. Furthermore, there was no proportion of unknown nucleotides (N percentage) in any of the studied libraries. The raw read numbers ranged from 44,378,972 to 46,206,808. The clean reads ranged from 43,537,934 to 44,853,794, with percentages of 98, 96, 97 and 97% for CR1, CR2, CR3 and CR4, respectively. The GC percentages were between 39 and 42% in the four libraries, as shown in Table 1.

### 2.2. Functional Annotations

To identify the functions of novel date palm genes, the differentially expressed genes (DEGs) were annotated with a BLASTX search against the NR NCBI protein database content with a cutoff E-value ≥10^−5^ depending on sequence similarity. In this study, 26,091 unigenes were detected, including 17,003 (65.17%) with uncharacterized proteins or unknown function. Furthermore, out of the total obtained genes, the sequences of 4646 genes were subjected to Gene Ontology (GO) analysis to determine the fundamental functions of the novel genes. A total of 8643 (33.12%) obtained sequences were submitted to the Clusters of Orthologous Groups/euKaryotic Orthologous Groups (COG/KOG) database, which revealed that the highest number of genes were categorized as “posttranslational modification, protein turnover and chaperones”. In addition, 8294 (31.79%) sequences had matches in the Kyoto Encyclopedia of Genes and Genomes (KEGG) database, 26,021 (99.73%) had matches in the Protein families (Pfam) database, and 25,136 (96.34%) matched to NCBI nonredundant (Nr) protein sequences (Figure 1).

### 2.3. Gene Ontology (GO) Analysis

The obtained sequences that were subjected to GO were classified into three categories: cellular components, biological processes and molecular functions. The genes in the cellular component category, which is related to cellular compartments and structures affected by UV-B included genes distributed in 15 subcategories. The cellular components most affected by UV-B were cell parts, organelles and membranes. This indicates that the genes responsible for different cellular components and organelles as well as membranes are highly affected by UV-B. Among genes categorized as being associated with biological processes, cellular processes and metabolic processes were the most enriched subcategories. For molecular function, the functions most affected by UV-B treatment were binding and catalytic activity, as shown in Figure 2.

### 2.4. Differentially Expressed Genes (DEGs)

Different expression patterns of 60-day-old date palm seedlings under UV-B stress were profiled to develop a better understanding of the plant response to UV light. After the seedlings were subjected to 72 h of UV-B treatment, differentially expressed genes, including those whose expressions were up- or downregulated, were compared to those of the control, the results of which are shown in Table 2. Among the four libraries, due to UV-B stress, compared to genes of the control, the expression of 10,249 and 12,426 genes was upregulated and downregulated, respectively. These differentially expressed genes are thus associated with UV-B stress.

Among these genes, whose expressions were upregulated, three were expressed at levels 8-fold higher than those in untreated plants, including those encoding the protein YLS9-like, germin-like protein 3-8 and the protein P21-like. The proteins encoded by these three genes whose expressions were upregulated act as plant cell defenders that activate the hypersensitive response against stress. These results indicate that genes play a major role in the response of plants to UV-B stress. Furthermore, there were two identified genes that were expressed in leaf tissues and whose expression was more than 7-fold higher in UV-B-treated plants than in the control plants. These two genes encoded the protein sensitive to proton rhizotoxicity 1-like and SNF1-related protein kinase regulatory subunit beta-1-like isoform X1, which regulated several mechanisms related to specific stress tolerance mechanisms, particularly H^+^ tolerance mechanisms.

Six genes were the third most highly expressed genes, with more than six-fold higher expression in UV-exposed leaf tissue than in control plant leaf tissue. These transcripts were aligned to pathogenesis-related protein 1-like, the transcriptional corepressor LEUNIG, auxin-induced protein X10A, glutathione S-transferase, and premnaspirodiene oxygenase-like, and there was one uncharacterized protein sequence. Furthermore, seven genes were expressed at levels fivefold-higher than those of the control; these genes have functions strongly related to responses to biotic and abiotic stresses (UV light; Table 2).

However, due to UV-B light, some plant functions were negatively influenced by the relatively low expression values of related genes. These genes included those that encoded galactosyltransferase 2; flavanone 3-hydroxylase; fatty acyl-CoA reductase 4; caffeoyl shikimate esterase-like; the protein LSD1 isoform X2, whose expression was seven-fold lower than that in control plants; the protein LSD1 isoform X2, whose expression was six-fold lower than that in the control. In addition, the expressions of eight transcripts were downregulated in the treated plants, the level of which was five-fold lower than that in the control. These sequences were aligned to CASP-like protein, an abscisic acid receptor, WAT1-related protein, 3-ketoacyl-CoA synthase, histone-lysine, flowering-promoting factor 1-like protein 3 and short-chain dehydrogenase reductase 3a isoform X2.

### 2.5. KEGG Pathways Related to UV Radiation

In response to UV-B stress, several pathways related to metabolic process regulation changed, showing many genes whose expression was up- and downregulated. Furthermore, the expression of various amino acid metabolic pathway-related DEGs dramatically increased after date palm was treated with UV-B irradiation. There were more DEGs involved in phenylalanine, tyrosine and tryptophan, as well as arginine and proline metabolism, than DEGs involved in other pathways, as shown in Figure 3. Additionally, there were more DEGs involved in phenylalanine metabolism, which is neatly related to secondary metabolism, than DEGs involved in most other metabolic pathways under UV-B treatment. These results indicated that UV-B irradiation has been shown to activate amino acids and flavonoid metabolism in date palm leaves. In this work, the identified metabolites, including amino acids and secondary metabolites, were performed comprehensive metabolic profiling to investigate the variations of primary and secondary metabolites in date palm leaves under UV-B irradiation. We identified/annotated 50 metabolites in date palm leaves during the course of UV-B irradiation by using an LC/MS-MS. The log2 fold changes in amino acid contents were determined by hierarchical cluster analysis (HCA). The highest levels of amino acids, including phenylalanine, tryptophan, valine, arginine, and tyrosine, were compared with those of the control, the results of which are shown in Figure 4. The results suggested that UV-B radiation might activate the biosynthesis of amino acids in date palm leaves, and these amino acids might be used to synthesize other metabolites that respond to UV-B radiation or might play important roles in protecting plants against UV-B stress. The results also showed that amino acids work as precursors that act as intermediates for secondary metabolite synthesis. Among the secondary metabolite synthesis pathways, shown in Figure 5, phenylpropanoid metabolism and flavonoid biosynthesis, including many metabolites, showed significant changes in response to UV-B irradiation compared to those in the control, as shown in Figure 6.

### 2.6. Transcription Factors

According to the NR database, the members of some families of transcription factors showed significant changes in gene expression levels. The most pronounced TF families included the F-box protein (10 and 4 genes, whose expression was upregulated and downregulated, respectively), MYB transcription factor (seven and one genes whose expressions were upregulated and downregulated, respectively), WRKY transcription factor (two and one genes whose expressions were upregulated and downregulated, respectively), chaperone protein (five and one genes whose expressions were upregulated and downregulated, respectively), protein NRT1/PTR (one and four genes whose expressions were upregulated and downregulated, respectively), and ABC transport (three genes each whose expressions were upregulated and downregulated) families, followed by the GTP binding (one gene each whose expressions were upregulated and downregulated) family. In addition, the expressions of some genes were significantly upregulated, such as those encoding NACs, bHLHs, protein kinases and calmodulin, with total numbers of upregulated genes of three, three, six and three, respectively, as shown in Table 3.

### 2.7. Chloroplast-Related Genes as Affected by UV Stress

Plants utilize energy from sunlight to perform photosynthesis in chloroplasts. To evaluate the UV-B stress effects on date palm chloroplast-related genes, differences in genes whose expression was up- or downregulated, and that differed from those of the control, were recorded. The chloroplast-related genes whose expression was most upregulated were those encoding zeaxanthin epoxidase, chaperone protein dnaJ-11 and protein kinase; their expression level was three times higher than that of the control. The expression of 12 genes was 2-fold higher than that of the control, as shown in Table 4. The obtained results showed that the genes whose expression was upregulated were mostly involved in increasing photosynthetic efficiency and in the photosystem II complex in addition to balancing redox reactions in chloroplasts and the response to changing light conditions. On the other hand, the detected genes whose expression was downregulated were involved in functions such as magnesium chelatase catalysis, contributions to the control of energy distribution between the two photosystems (PSI and PSII) and the reassimilation of ammonia generated by photorespiration.

### 2.8. Photosynthesis

Photosynthesis is the mechanism in which energy from the sun is transformed into chemical energy in sugars. Furthermore, photosystems play a key role in light reactions through large complexes of proteins and pigments (light-absorbing molecules) that react to and harvest light energy. In the current study, great variations occurred in photosystem reactions (PSI and PSII) due to UV-B exposure. Among all genes related to photosystem mechanisms, there were 6 and 7 genes in PSI and PSII, respectively, whose expression changed compared to that in the control. These genes were involved in the PsbO, PsbP, PsbQ, PsbR, PsbS, PSB27-H1 and Psb28 elements in PSII. In addition, another six elements involved genes whose expression was downregulated, viz., PsaD, PsaK, PsaG, PsaL, PsaN and PsaO, in PSI. Photosynthetic electron transport (PET) converts free and abundant sunlight energy into reducing power and chemical energy by transferring electrons in PSI and PSII in chloroplasts. PET is a process that occurs at four different sites: plastocyanin (PC), ferredoxin (FD), ferredoxin NADP(H) oxidoreductase (FNR) and cytochrome c6 (Cyt-c6). Among these four sites, three included two genes whose expressions were downregulated (PC and FD) and one gene whose expression was upregulated (FNR). FNR has a main role in catalyzing the final step of PET, providing NADPH for CO_2_ assimilation and another reductive metabolism. However, Cyt-c6 genes did not show a significant effect due to UV-B stress compared that in the control plants, as shown in Figure 7.

### 2.9. Oxidative Phosphorylation-Related Genes

Oxidative phosphorylation is the mechanism by which energy is harnessed to synthesize ATP via a sequence of protein complexes embedded in the inner membrane of mitochondria. In this study, the expression of genes related to oxidative phosphorylation was significantly affected by UV stress; an intramitochondrial signaling pathway that regulates cytochrome oxidase (COX) was also affected. COX also contributes to energy storage in the form of an electrochemical gradient used for the synthesis of ATP via the oxidative phosphorylation system. In addition, the cytochrome C reductase fbcH gene product, which is synthesized as a polyprotein precursor for cytochromes b and c1, showed a pronounced effect, in terms of its gene expression (Figure 8).

### 2.10. Quantitative Real-Time PCR (qRT-PCR) Validation of DEGs from RNA-seq

A validation experiment was conducted using qRT-PCR to confirm the accuracy of the Illumina high-throughput sequencing data. Ten unigenes are chosen at random for quantitative RT-PCR assays: LOC103699391, LOC103695855, LOC103709833, LOC103711915, LOC103715616, LOC103707665, LOC103715998, LOC103719605, LOC103702656 and LOC103716074. The qRT-PCR results showed that the expressions of all of the tested genes were upregulated after treatment with UV-B irradiation, which was compliant with RNA-seq data (Figure 9). Thus, the RNA-seq results were accurate for the identification and measurement of the expression of DEGs involved in different processes in date palm leaves in response to UV-B radiation.

## 3. Materials and Methods

### 3.1. Plant Materials

Sixty-day-old *Phoenix dactylifera* cv. Khalas plants (first true-leaf stage) were collected by Egyptian Desert Research Center staff from green valley, Al-Kharga with latitude 25°27′16.3476″ N 30°33′12.1608″ E and were treated with UV-B (wavelength, 253.7 nm; intensity, 75 μW cm^−2^). The UV-B full lamp description is as follow; a UV-B chamber (Philips, Netherlands, TL8W/302 nm narrowband UVB tube; 12.8 µW cm^–2^; measured using a UV radiometer with the UV-295 detector from photoelectric instrument factory of Beijing normal university) was used in this experiment. There was no filter between the lamp and the treated plants [28]. The seedlings were subjected to UV-B stress for 72 h (16 h of UV light and 8 h of darkness). After the treatment, no visible symptoms appeared during the radiation treatment. Samples from the first true leaves were collected from the control as well as UV-B-treated seedlings and directly frozen in liquid nitrogen for RNA extraction.

### 3.2. RNA Extraction

Total RNA was extracted from collected samples by using the TRIzol method (Life Technologies, Carlsbad, CA, USA) according to the manufacturer’s protocol. The RNA sample purity was determined by using a Nanophotometer^®^ spectrophotometer (IMPLEN, Westlake Village, CA, USA). To detect the integrity and concentration of RNA samples, an Agilent 2100 RNA Nanodrop 6000 was used (Agilent Technologies, Santa Clara, CA, USA).

### 3.3. CDNA Library Preparation for Transcriptome Analysis

The total amount of RNA for second-generation sequencing was 3 μg, which was used as input material for the preparation of the library. Sequencing libraries were generated using a NEBNext^®^ Ultra™ RNA Library Prep Kit for Illumina^®^ (NEB, Ipswich, MA, USA), following the manufacturer’s protocol, to assign sequences to each sample, index codes were added. Briefly, purified mRNA was attached to magnetic beads using Poly-T. oligo. After the libraries were constructed, PCR products were purified, and library quality was assessed on the Agilent Bioanalyzer 2100 system.

### 3.4. Data Analysis Quality Control

The quality of the raw data (raw reads) in fastq format was processed through in-house Perl scripts. After we removed low-quality sequences and sequences contaminated with adapters, all subsequent analyses were based on clean reads. The raw reads sequenced from the Illumina platform were processed to obtain high-quality sequences (clean reads). The Q30 value, the GC content and the clean data sequence duplication level were determined simultaneously. All downstream analyses were based on clean, high-quality data.

### 3.5. Transcriptome Based Assembly

TPM, FPKM, RPKM and fold change (FC) values were recorded for each replicate of each library separately. The acquired sequence was aligned, and comparable sequence data were obtained from all libraries [29]. Trinity was used for further analysis [29], with min_ kmer_cov set to 2 by default and with all other parameters set to the default.

### 3.6. Gene Annotation and Functional Analysis

To conduct a functional gene annotation, different source databases were used, including the NCBI, Pfam, KOG/COG and GO databases. NCBI nonredundant protein sequences (Nr) (https://www.ncbi.nlm.nih.gov/ (accessed on 3 February 2021)) with an e-value cutoff of 1E-5, Nt sequences with an e-value cutoff of 1E-5, Pfam (Protein families) (https://pfam.sanger.ac.uk/ (accessed on 3 February 2021)) sequences with an e-value of 1E–2 and KOG/COG (Clusters of Orthologous Groups of proteins) sequences with an e-value of 1E–3 were used. With respect to GO (Gene Ontology) annotations (http://geneontology.org/ (accessed on 3 February 2021)), there are three total ontology categories: Those that identify gene-related molecular functions, cellular components, and biological processes [30]. Our transcriptome was constructed and analyzed by Annoroad Genome Company methods (www.genome.cn (accessed on 3 February 2021)), and all the reference data were downloaded and analyzed in the period between 23 January 2019 to 23 April 2019. The methods were as follows: Alignment of the reference genomes, and the annotation file was downloaded from ENSEMBL database (http://www.ensembl.org/index.html (accessed on 3 February 2021)). Bowtie2 v2.2.3 was used for building the genome index, clean data and aligned to the reference genome using HISAT2 v2.1.0. HISAT2 is the successor to TopHat2, which uses a modified BWT algorithm to convert reference genomes to the index for faster speed and fewer resources. For Function Enrichment Analysis, the GO (Gene Ontology, http://geneontology.org/ (accessed on 3 February 2021)) enrichment of DEGs was implemented by the hypergeometric test, in which the p-value is calculated and adjusted as a q-value, and the data background comprises genes in the whole genome. GO terms with q. GO enrichment was introduced for the differentially expressed genes (DEGs) by the hypergeometric distribution method [31]. GO enrichment was considered significant when the threshold q < 0.05 was met. Significant GO functional enrichment analysis can determine the main biological functions associated with differentially expressed genes.

The KEGG (Kyoto Encyclopedia of Genes and Genomes) database [32] is a resource for genome deciphering (https://www.genome.jp/kegg/ (accessed on 3 February 2021)). Significant enrichment analysis of each pathway in the KEGG database was implemented by the hypergeometric test method.

### 3.7. Quantification of Levels of Gene Expression and Study of Differential Expression

For each sample plant, the gene expression levels were calculated using RSEM 30. Clean data were mapped back to the assembled transcriptome. Read counts were obtained from the mapping results for each gene in all samples counted by HTSeq v0.6.0 on the basis of FPKM (fragments per kilobase million mapped reads). The DEGseq2 R package reported in [33] was used for determining the differential expression among the digital gene expression data using a model based on a negative binomial distribution. To identify genes whose expression significantly differed, FDR ≤ 0.05 and a (|log2(fold change (FC))|) ≥ 1 were set as the thresholds for significance (DEGs) [33].

### 3.8. Quantitative and Real Time-PCR (qRT-PCR) Validation

To check the precision of the Illumina sequencing results, the different expression levels of the DEGs were confirmed via qRT-PCR using 10 random unigenes. Total RNA was extracted from collected samples with the TRIzol method (Life Technologies, Carlsbad, CA, USA), reverse transcribed (Takara, http://ww.takara-bio.com/ (accessed on 3 February 2021)), and subjected to PCR in a BioRad CFX96 Real-Time System according to the manufacturers’ instructions. RT-qPCR was carried out using the technique described in [34]. The specific primers used for the ten genes were designed using Primer 5 software (http://www.ncbi.nlm.nih.gov/tools/primer-blast/primer3 (accessed on 3 February 2021)). In triplicate, all the reaction solutions were prepared and relative expression was calculated using the 2^(-Ct) method, with the expression normalized against the internal reference gene (Tubulin) [35]. Sequences of the primers used and the identities of the BLAST hits are provided in Table 5.

### 3.9. Metabolite Analysis by LC-MS/MS

Samples of leaves were analyzed through an HPLC-ESI-QTOF-MS/MS system (6520B, Agilent, Santa Clara, CA, USA), and fragmentation patterns were obtained in targeted MS2 mode. The data were processed using MassHunter Qualitative Analysis software (Agilent Technologies, Barcelona, Spain) [36]. Multiple reaction monitoring (MRM) was performed using an LC-ESI-Q TRAP-MS/MS (4000Q TRAP, ABI, Framingham, MA, USA) for the quantification of the metabolites. Data acquisition, curve calibration, peak integration, and calculations were performed with Analyst 1.6 software (AB SCIEX). The data processing and the analytical conditions were similar to those previously mentioned [36]. The qualitative and quantitative chromatographic parameters were the same: The HPLC column was a Shim-pack VP-ODS C18 (pore size 5.0 μm, length 2 × 150 mm); the column temperature was set at 40 °C; the solvent system was water (0.04% acetic acid added): acetonitrile (0.04% acetic acid added). This method was used according to [36]. To quantify the amino acids and flavonoids, the area of each individual peak was calculated and compared to those of the standard curves, as reported previously [34,36].

Statistical analyses. Principal component analysis (PCA) was performed with R (www.r-project.org/ (accessed on 3 February 2021)) using the Pareto scaling method to obtain grouping information for nontargeted metabolomics data. Seventeen amino acids and 36 flavonoids were subjected to hierarchical clustering analysis via HemI (http://hemi.biocuckoo.org/ (accessed on 3 February 2021)) to visualize the changes in metabolite profiles.

## 4. Discussion

In this study, HiSeq Illumina sequencing was used to characterize the transcriptome profiles of 60-day-old date palm seedlings in response to UV-B stress. Transcriptome analysis revealed a total of 44,853,794 clean reads, which were assembled into thousands of predicted genes. In total, 10,249 and 12,426 genes, whose expressions were upregulated and downregulated, were detected, compared to those in the control. A total of 96.34% of the identified unigenes were submitted to the NR protein database, as shown in Figure 1.

The current results revealed that the most affected cellular components involving the largest number of genes according to GO analysis were cell parts, organelles and membranes. This indicates that the plants try to protect their cells and organelles from UV-B exposure. In addition, the cellular process and metabolic process subcategories ranked first among the biological processes that help the plant produce antioxidants to protect biological process cycles against oxidative stress caused by UV-B radiation. For molecular function, the functions most affected by UV-B treatment were binding and catalytic activity. The balance between catalytic and metabolic processes determines the degree of resistance of plant cells to excess UV-B stress, which affects overall plant status. Our results agree with those of a previous report [34,37].

Gene expression analysis and functional annotation were performed on the genes overexpressed in response to excess UV-B. The genes whose expression was most upregulated included those encoding YLS9-like, germin-like protein 3-8 and P21-like. The products of these groups of genes are involved in ultraviolet radiation absorption, which is related to the plant defense system that activates the hypersensitive response against UV stress. Germin-like proteins (GLPs) can also increase the expression level of radiation resistance-related genes, which enhance plant tolerance to different adverse environmental stresses, including excess radiation [38]. This hypersensitive action protects plant organelles against UV-B stress, which explains why the extremely high expression of these genes was significantly upregulated in date palm. Furthermore, proteins sensitive to proton rhizotoxicity 1-like and SNF1-related protein kinase regulatory subunit beta-1-like isoform X1 regulate many mechanisms related to specific stress tolerance mechanisms, particularly H^+^ tolerance mechanisms. Furthermore, it was found that the harmful effects of UV-B radiation increased under combined stresses, especially at low pH. This explains the high expression level of these two genes in leaf tissues (seven-fold higher than that in untreated plants). These two genes contribute to maximizing the plant defense mechanism against UV stress. In this study, UV-B radiation was found to influence multiple biological processes in plants either through various regulatory effects or via direct damage, and these results are in agreement with those in [39].

In addition, a group of genes were highly expressed as a result of UV-B stress, including pathogenesis-related (PR) protein 1-like, the transcriptional corepressor LEUNIG, glutathione-*S*-transferases (GSTs) and premnaspirodiene oxygenase-like. PR proteins play crucial roles in the plant defense system [40], and the transcriptional corepressor LEUNIG has a role in regulating genes that are involved in a variety of physiological processes, including disease resistance, response to DNA damage, and cell signaling. [41]. The detoxification of xenobiotics, limiting oxidative damage and other stress responses in plants have been associated with GSTs [42]. Premnaspirodiene oxygenase-like mainly participates in the synthesis of lignin, UV protectants, pigments, defense agents, hormones, fatty acids, and molecules for signaling [43]. Therefore, these genes increase tolerance to UV-B stress. As in the current study, a previously conducted study proved that these groups of genes have biological functions strongly related to responses to biotic and abiotic stresses, as well as UV light.

However, the current results found that the expression of the galactosyltransferase 2, flavanone 3-hydroxylase, fatty acyl-CoA reductase 4, caffeoyl shikimate esterase-like and protein LSD1 isoform X2 genes was downregulated compared to that of the control.

In this study, we analyzed the transcript levels of unigenes in the transcriptome data related to metabolic processes that were affected by UV-B radiation. We determined the nontarget metabolite contents and analyzed the metabolites that had significantly different levels compared with those of the control to explore the relations between UV-B radiation and metabolic processes. In this study, we identified many unigenes involved in different pathways in response to UV-B stress, and the shifts in primary and secondary metabolites were investigated [34].

The results showed that the unigenes related to the amino acid, phenylpropanoid and flavonoid biosynthesis pathways were affected by UV-B radiation. Therefore, the changes in metabolite contents were related to the difference in the unigenes transcript levels involved in metabolite pathway responses to UV-B stress. Amino acids are monomers that play major roles in forming proteins and are a major primary metabolite in plants. Additionally, amino acids can be catabolized into intermediates involved in many metabolic processes, such as the TCA cycle [34,44,45]. Amino acids play a wide variety of roles in protecting plants against different stresses and act as precursors of secondary metabolites related to plant defense against UV-B radiation [46]. It was found in previous studies that many amino acids, including glutamate, isoleucine, leucine, serine and proline, accumulate in plants affected by exposure to high levels of UV-B irradiation [34,47]. In addition, amino acid metabolism-related pathway activity was shown to increase or be induced, such as those involving phenylpropanoids and flavonoids (secondary metabolites), according to [34,48]. In this study, variations in amino acid (lysine, phenylalanine, tyrosine, glutamate, leucine, proline and alanine, etc.) contents in response to UV-B radiation were found in date palm leaves. Most of the amino acids increased significantly Compared with the control, under UV-B irradiation treatment, which revealed that date palm leaves tended to accumulate amino acids when exposed to UV-B radiation, as shown in Figure 3. Lysine increased compared with the control under UV-B irradiation treatment, and this result revealed that lysine responded to UV-B irradiation, which is consistent with a previous study, showing that lysine is involved in plant stress responses [34,44]. Lysine might be an intermediate of precursors of the TCA, as lysine is concerted into the energy-associated TCA cycle metabolite acetyl-CoA [49] in response to energy loss under conditions of stress [44]. According to these results, Under UV-B irradiation, lysine may play an important role in stress response. Glutamate levels decreased after 48 h of UV-B irradiation compared to the control levels, and decreased levels of glutamine play a major role in preserving the redox state by conversion into GABA and GSH in date palm leaves under UV-B irradiation, as shown in previous studies [34,50]. This result showed the accumulation of glutamine and GABA in response to UV-B stress, and GABA reduced oxidative stress caused by UV-B irradiation, which is consistent with the findings in previous studies [34]. In addition, some amino acids, including aspartate, lysine and glutamine, are closely associated with the formation of a regulatory metabolic pathway in date palm leaves in response to UV-B irradiation by forming antioxidant compounds. These results are in agreement with those of a previous study [34]. Furthermore, some amino acids contributed to the biosynthesis of secondary metabolites, such as flavonoids, under UV-B irradiation [34,51]. Phenylalanine and tyrosine participate as intermediates in the phenylpropanoid pathway. This leads to flavonoid biosynthesis in plants under UV-B irradiation. [52]. These results revealed that phenylpropanoids are involved in the biosynthesis of flavonoids, which help to protect date palm against UV-B stress. Furthermore, flavonoids accumulate in plants under different stresses, including UV-B stress, and function as inhibitors of ROS generation and as ROS scavengers [20,21,53,54]. In this study, the contents of many flavonoid under UV-B irradiation increased compared with those of the control, as shown in Figure 4.

In the current study, transcription factors were identified as regulatory proteins that are involved in regulating the expression of other genes that participate in the UV-B stress response, including F-box, MYB, WRKY, chaperone, NRT1/PTR family member, ABC transporter, GTP-binding, NAC, bHLH, kinase and calmodulin proteins. These transcription factors are expressed in response to ultraviolet absorption. They are triggered by various signal transduction pathways and can bind to cis-acting elements directly or indirectly to modulate the transcription efficiency of target genes. Therefore, these proteins can act as key regulators of crop genetic improvement.

Furthermore, members of the MYB transcription family were found to have a positive effect in synthesizing phenylpropanoid compounds to absorb UV-B radiation and that are involved in regulating the balance for the production and accumulation of sunscreen-type compounds for radiation tolerance (Jin, H. et al., 2000) [8]. These transcription factors enhance tolerance to UV-B radiation and pathogen attack through enforcing the walls of plant cells to increase the accumulation of wax and deposition of lignin [55].

Chloroplast genes are related to light absorption and have an important role during exposure to UV-B stress. In the present study, the chloroplast-related genes whose expressions were most upregulated included zeaxanthin epoxidase, chaperone protein dnaJ-11 and protein kinase, presenting expression levels three times higher than those of the control. Zeaxanthin epoxidase is related to carotenoids and has an essential function in the photosynthetic machinery. In addition to its important role in light harvesting, this enzyme also plays an important role in de-epoxidation via the inhibition of photosynthetic activity under UV-B stress [56]. In the present experiment, protein kinase was found to be hyperactivated in response to UV-B, which is photo-repair deficiency, indicating that UV-damaged DNA is a cause for protein kinase signaling [57].

It was found that there was a total of 12 genes expressed in date palm leaf tissues, with levels more than two-fold greater than those in the control. These genes are involved in increasing plant defense against conditions of environmental stress, such as high light, H_2_O_2_, methyl viologen and copper sulfate. Furthermore, copper chaperones for superoxide dismutase play a major role in protecting plants against oxidative stress caused by UV-B exposure [58]. Cyanidin 3-O-rutinoside 5-O-glucosyltransferase-like is responsible for accumulating cyanidin 3-O-rutinoside in large amounts under UV-B stress. Systematically, it has a major role in effective protection against UV stress [59]. In agreement with the current results, it was found that the most pronounced function of these genes is reducing oxidative damage and UV light penetration into the layers of photosynthetic cells; moreover, it increases DNA repair and antioxidative defense [60].

In this study, chloroplast genes whose expressions were downregulated included thioredoxin M-type protein, early light-induced protein 1, magnesium-chelatase subunit ChlH, CURVATURE THYLAKOID 1B, anthranilate synthase beta subunit 2, photosystem II 22 kDa protein, early light-induced protein, photosystem II 10 kDa polypeptide, phosphoglycolate phosphatase 1B and a glutamine synthetase leaf isozyme. The downregulated expression of thioredoxin M-type due to UV-B stress was found to reduce the stability of the photosystem II complex and decrease reactive oxygen species levels. It also increases oxidative activity, which can affect plant mechanisms, according to [61]. Early light-induced protein 1 is involved in the regulation of the redox state of the cell and plays an important role in protecting the photosystem under photo-oxidative stress. In this study, the expression of early light-induced protein 1 was downregulated in response to UV-B stress. The downregulated expression of these genes negatively affects the function of protecting the photosystem under photooxidative stress [62]. Depending on the abovementioned findings, it can be said that, under UV-B stress, the chloroplast genes whose expression was downregulated caused the misregulation of the redox state of the cell and protection of the photosystem, which negatively affected plants.

Photosystem mechanisms involve three processes: photosystem II, photosystem I and photosynthetic electron transport. The effects of UV-B stress on Photosystem II induced the downregulated expression of many genes, including Psbo, Psbp, PsbR, PsbQ, PsbS, Psb27 and Psb28. In addition, the water-splitting and oxygen-evolving reactions catalyzed during the photosynthetic process by Photosystem II were highly affected. UV-B radiation damage affects the catalytic Mn cluster involved in water oxidation, which is most likely sensitive to the UV absorption of MN(III) and MN(IV) ion ligation by organic residues, according to [63]. Furthermore, Photosystem I was also affected by UV-B, which downregulated the expression of several genes, including PsaD, PsaG, PsaK, PsaL, PsaN and PsaO. PSI and PSII can physically dissociate from the major mobile light-harvesting chlorophyll a/b complex II (LHCII), according to [64]. The photosynthetic electron transport process was associated with genes whose expressions were downregulated, including petE (PC), Fd (petF) and FNR (PetH). Furthermore, the expression of most photosynthesis-related genes was downregulated in response to UV-B stress. In the current study, UV-B significantly reduced the activity of photosystem I, photosystem II and the whole chain. These results are in agreement with those of [65]. In addition, the expressions of most oxidative phosphorylation related genes were up-regulated, i.e., cytochrome oxidase (COX10 and COX6B), was upregulated. In this study, it was found that UV-B radiation mediated oxidative stress as a protective mechanism by producing ROS and stimulating the activity of antioxidant enzymes, which was also previously reported by [66].

The results of qRT-PCR corresponded with the expression patterns of unigenes detected in the RNA-seq data. The upregulated expression of these genes suggests that various pathways in date palm leaves were caused by UV-B irradiation. We analyzed the transcriptome data to investigate the differentially expressed genes related to UV-B stress and compared the genes whose expression was upregulated and downregulated, which were shown in our results (transcription factor-related genes, chloroplast-related genes and photosystem-related genes) under UV-B stress. Furthermore, comprehensive metabolic profiling was performed to explore the variations in primary and secondary metabolites in date palm leaves under UV-B irradiation. We identified/annotated 50 metabolites in date palm leaves under UV-B irradiation by using an LC-MS/MS system. The altered levels of various metabolites, mainly those involved in the metabolism of amino acids and pathways of flavonoid biosynthesis, together with the variation in associated unigenes in the transcript stages, shed light on the potential interactions of amino acid metabolism with plant energy activities and secondary metabolism in date palm leaves in response to UV-B irradiation. Our findings enhance the understanding of the metabolic control of date palm leaves in response to UV-B irradiation and pave the way for further dissection under stress conditions of metabolic pathways in date palm.

## Figures and Tables

**Figure 1 ijms-22-02564-f001:**
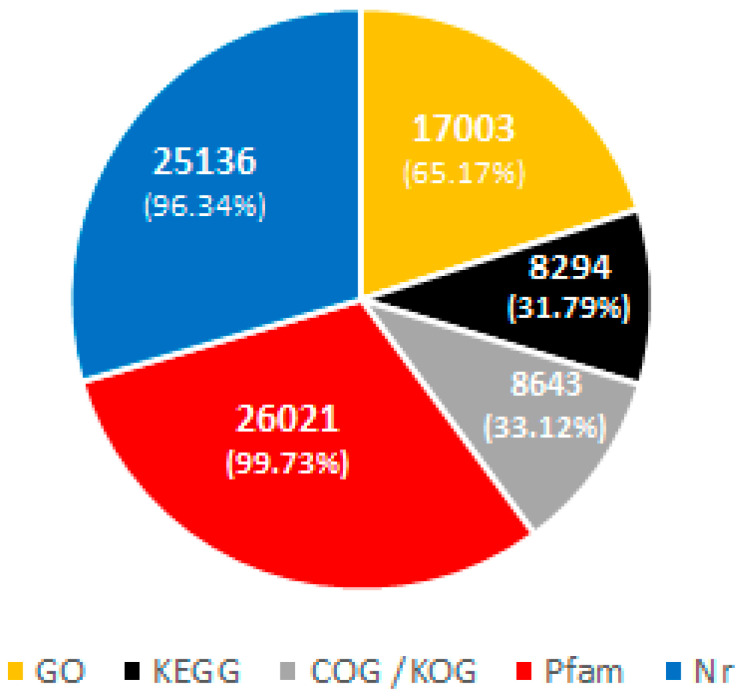
Annotated number Values (%) GO, KEGG, COG/KOG, pfam and NR percentage.

**Figure 2 ijms-22-02564-f002:**
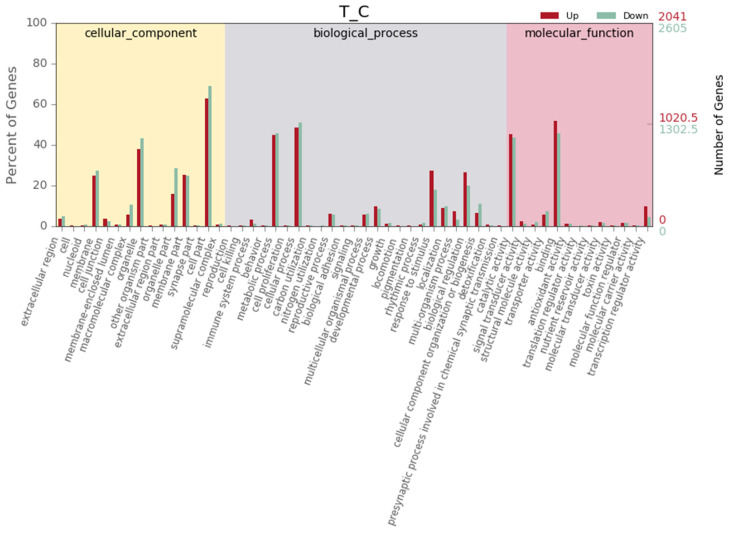
GO functional classification of Date palm unigenes. Gene Ontology (GO) terms are summarized in three main categories of biological process, molecular function and cellular component.

**Figure 3 ijms-22-02564-f003:**
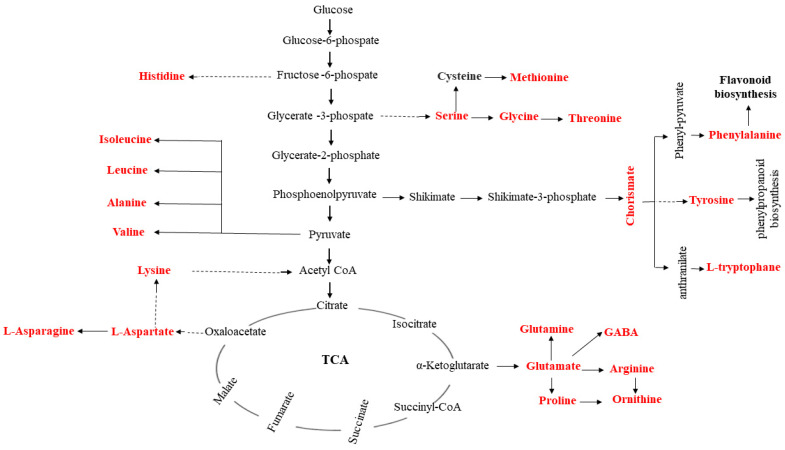
Changes in amino acid levels in the context of amino acid biosynthesis pathways in date palm leaves under UV-B irradiation. Metabolites detected shown in red color, solid lines represent one-step reactions, and dashed lines represent multi-step reactions.

**Figure 4 ijms-22-02564-f004:**
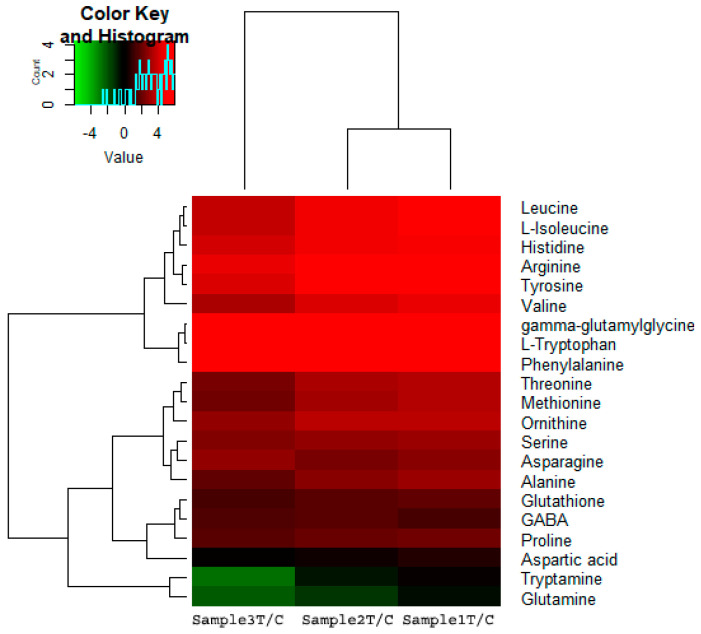
Variations in amino acids levels in date palm leaves under UV-B irradiation compared to the controls. The log2 (fold change) value of three samples treated compared to the control was normalized to complete hierarchical clustering analysis. Red indicates higher amino acid contents in treated samples than in the control, whereas green indicates lower amino contents in treated samples compared to the controls.

**Figure 5 ijms-22-02564-f005:**
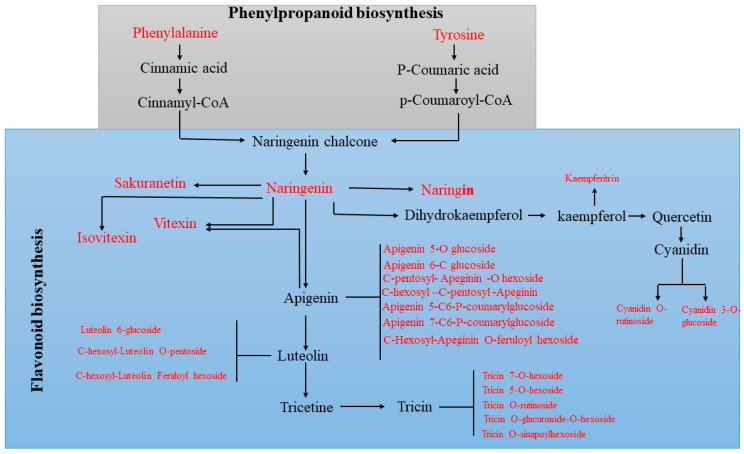
Modulation of phenylpropanoid and flavonoid biosynthesis pathways in date palm leaves under UV-B irradiation. Detected metabolites are indicated with red color.

**Figure 6 ijms-22-02564-f006:**
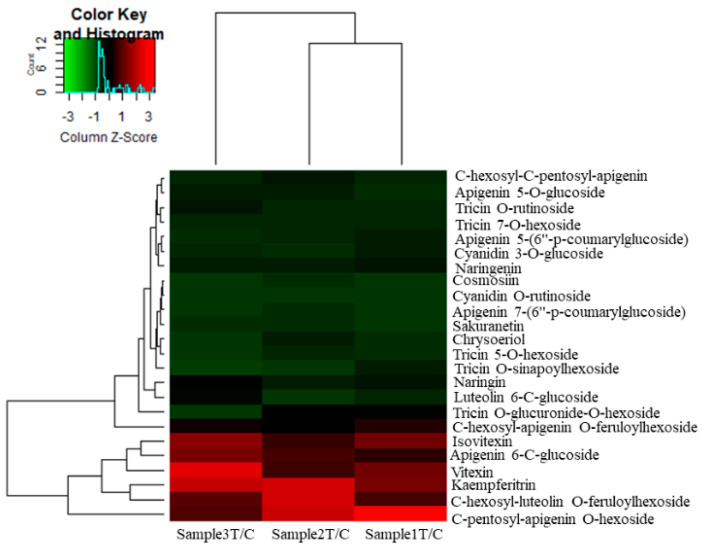
Variations in flavonoid levels in date palm leaves under UV-B irradiation compared to the controls. The fold change value of three samples treated compared to the control was normalized to complete hierarchical clustering analysis. Red indicates higher flavonoids contents in treated samples compared to the control, whereas green indicates lower flavonoids contents in treated samples compared to the controls.

**Figure 7 ijms-22-02564-f007:**
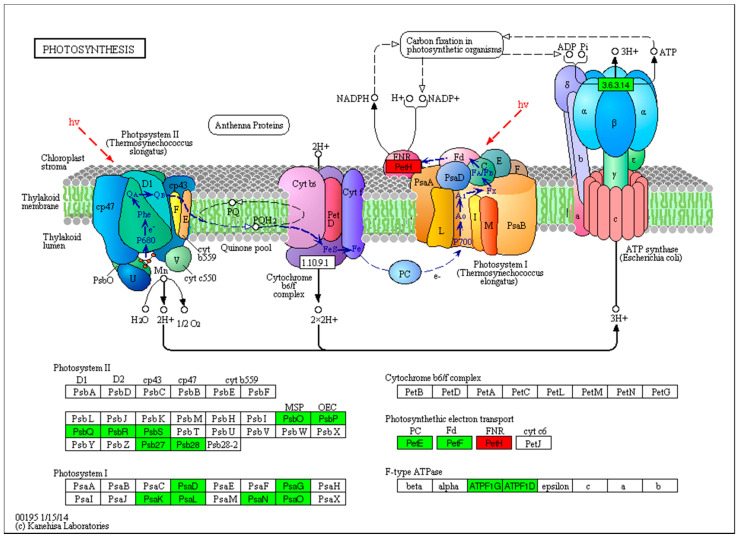
Photosynthesis related genes as affected by UV-B stress.

**Figure 8 ijms-22-02564-f008:**
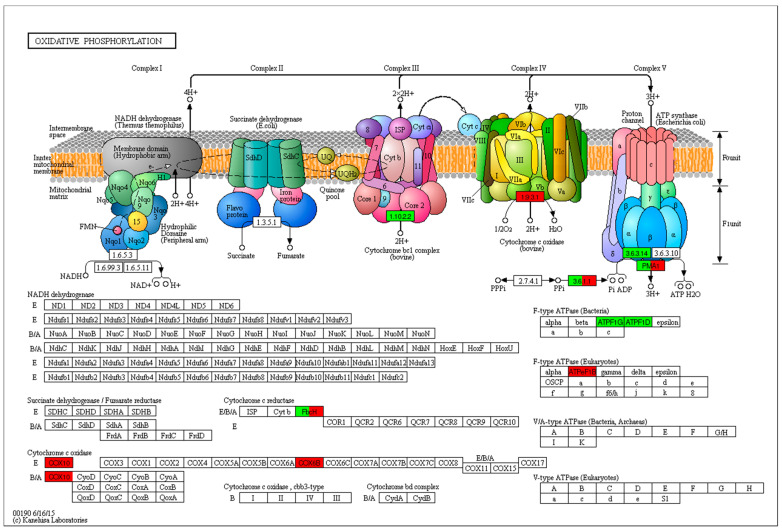
Oxidative phosphorylation-related genes as affected by UV-B stress.

**Figure 9 ijms-22-02564-f009:**
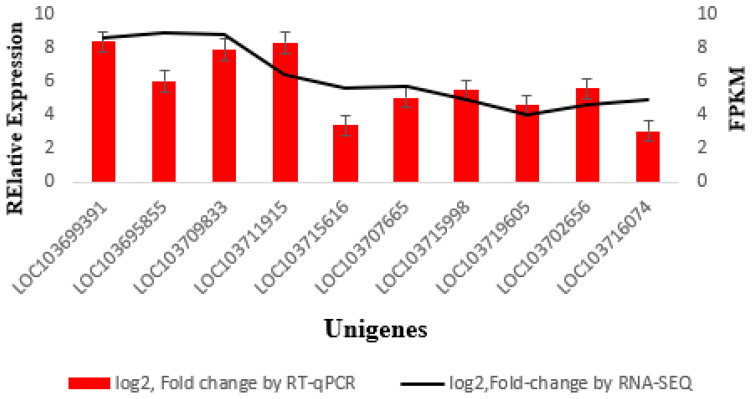
Quantitative real-time PCR analysis and fragments per kilo base million mapped read (FPKM) of 10 randomly chosen different expression genes DEGs, expression levels determined by qRT-PCR. Black line presented log_2_FPKM of RNA-seq data and red columns presented log_2_ the fold change of qRT-PCR.

**Table 1 ijms-22-02564-t001:** Transcriptome sequencing statistical summary of sequenced and assembled results.

	KhalasCR1	KhalasCR2	KhalasUR1	KhalasUR2
Total Reads	43,537,934	44,128,934	43,871,714	44,853,794
Mapped Reads	41,339,168	41,716,324	41,324,006	42,319,388
Unmapped Reads	2,198,766	2,412,610	2,547,708	2,534,406
Raw Bases Number	6,656,845,800	6,829,804,800	6,770,265,000	6,931,021,200
GC%	40%	39.7%	41.6%	39.9%
Q30%	94.59%	93.89%	93.05%	94.6%
Ns Reads (%)	0.15	0.17	0.26	0.27
Raw Reads Number	44,378,972	45,532,032	45,135,100	46,206,808
Clean Reads Number	43,537,934	44,128,934	43,871,714	44,853,794
Clean Reads Rate (%)	98.11	96.92	97.2	97.07
N percentage	00.00	00.00	00.00	00.00

Nt, total number of clean nucleotides; the GC% is the proportion of guanidine and cytosine nucleotides among total nucleotides; the Q30% is the proportion of nucleotides with a quality value >30, respectively; the N% is the proportion of unknown nucleotides in clean reads.

**Table 2 ijms-22-02564-t002:** Differentially expressed genes (DEGs) in date palm leaf tissues under UV-B stress.

Unigene ID	Log-Fold Change	Nr.annotation	Unigene ID	Log-Fold Change	Nr.annotation
Upregulated			Downregulated		
g28829	8.65	proteinYLS9-like	g12140	7.29	galactinol--sucrose galactosyltransferase 2
g25277	8.94	germin-like protein 3-8	g13825	7.36	flavonol synthase/flavanone 3-hydroxylase
g11369	8.79	protein P21-like	g16474	7.28	fatty acyl-CoA reductase 4
g7305	7.07	protein SENSITIVE TO PROTON RHIZOTOXICITY 1-like	g11969	7.62	caffeoylshikimate esterase-like
g23553	7.99	SNF1-related protein kinase regulatory subunit beta-1-like isoform X1	g15215	−6.21	protein LSD1 isoform X2
g20545	6.12	pathogenesis-related protein 1-like	g10248	−5.64	CASP-like protein RCOM_0864260
g29694	6.20	transcriptional corepressor LEUNIG	g11089	−5.21	abscisic acid receptor PYL2
g13273	6.40	auxin-induced protein X10A	g18749	−5.20	CASP-like protein 6
g11179	6.23	probable glutathione S-transferase	g3051	−5.75	WAT1-related protein At1g09380
g23192	6.33	hypothetical protein [Beta vulgaris subsp. vulgaris]	g17284	−5.89	3-ketoacyl-CoA synthase 6-like
g20792	6.91	premnaspirodiene oxygenase-like	g14976	−5.12	histone-lysine N-methyltransferase EZ1-like isoform X1
g8764	5.52	calcium-binding protein CML36	g17477	−5.85	flowering-promoting factor 1-like protein 3
g16674	5.59	glucosidase 2 subunit beta	g13339	−5.36	short-chain dehydrogenase reductase 3a isoform X2
g11260	5.61	strigolactone esterase D14	g17022	−4.28	delta (8)-fatty-acid desaturase 1-like
g27446	5.86	mavicyanin-like	g17881	−4.26	transcription repressor OFP13-like isoform X2
g6909	5.85	calcium-binding protein CML36	g27128	−4.95	probable beta-1,3-galactosyltransferase 6
g9368	5.73	ABC transporter B family member 11-like	g15595	−4.37	very-long-chain 3-oxoacyl-CoA reductase 1
g5355	5.45	putative receptor-like protein kinase At4g00960	g12155	4.45	homeobox protein BEL1 homolog
g20793	4.10	premnaspirodiene oxygenase-like	g4508	−4.02	protein IQ-DOMAIN 1
g29073	4.17	putative disease resistance protein RGA3	g10506	−4.91	transcription factor RF2b-like
g29590	4.99	cysteine-rich receptor-like protein kinase 10, partial	g734	−4.49	glucan endo-1,3-beta-glucosidase 14-like isoform X1
g28226	4.52	premnaspirodiene oxygenase-like	g967	−4.58	long chain acyl-CoA synthetase 6, peroxisomal-like
			g10070	−4.95	DNA-damage-repair/toleration protein DRT100

**Table 3 ijms-22-02564-t003:** Identified transcription factors in date palm leaf tissues under UV-B stress.

Unigene ID	Log-Fold Change	Nr.annotation	Unigene ID	Log-Fold Change	Nr.annotation
**F-box**					
g15721	3.04	F-box/LRR-repeat protein At4g14103-like	g10026	−1.99	F-box/kelch-repeat protein At5g60570-like
g16924	2.71	F-box/LRR-repeat protein 14 isoform X1	g124	−3.60	F-box/kelch-repeat protein At3g61590-like
g8642	2.21	F-box/LRR-repeat MAX2 homolog A-like	g10348	−1.27	EIN3-binding F-box protein 1-like
g24916	1.57	F-box protein At2g26160-like	g24	−5.99	F-box protein At5g49610-like
g14036	1.89	F-box/kelch-repeat protein At5g15710-like			
g12573	1.77	F-box protein At4g00755-like isoform X2			
g2920	1.08	F-box/kelch-repeat protein SKIP6-like			
g24233	1.79	F-box-like/WD repeat-containing protein TBL1XR1			
g2912	1.15	F-box/kelch-repeat protein At1g74510-like			
g6822	1.04	F-boxprotein At4g18380-like			
**MYB-transcription factor**					
g10079	4.47	myb-related protein Myb4	g25632	−3.89	myb-related protein 306-like
g7661	3.51	myb-related protein 315-like			
g21155	2.13	myb-related protein 306-like			
g26123	2.22	myb-related protein Zm1-like			
g15549	1.83	target of Myb protein 1-like isoform X1			
g6154	2.52	transcription repressor MYB5			
g9356	1.57	myb-like protein X			
**NAC transcription factor**					
g11223	4.88	NAC domain-containing protein 68-like			
g10695	3.92	NAC transcription factor 29-like			
g7306	1.07	NAC domain-containing protein 78			
**bHLH transcription factor**					
g19016	3.03	transcription factor bHLH51-like			
g25077	2.89	transcription factor bHLH30-like			
g21380	2.18	transcription factor bHLH79-like isoform X1			
**WRKY**					
g27377	3.41	WRKY transcription factor 9	g2524	−1.43	WRKY transcription factor 44
g10593	1.73	WRKY transcription factor 22-like			
**Protein kinase**					
g26059	3.06	protein kinase 2B, chloroplastic-like			
g22647	2.20	protein kinase APK1A, chloroplastic			
g24262	2.13	shaggy-related protein kinase alpha isoform X1			
g27267	2.25	shaggy-related protein kinase alpha-like			
g5427	1.42	Protein kinase APK1A, chloroplastic isoform X1			
g29590	4.99	cysteine-rich receptor-like protein kinase 10, partial			
**Chaperone protein**					
g3048	3.18	chaperone protein dnaJ 11, chloroplastic-like	g17272	−3.35	BAG family molecular chaperone regulator 1-like isoform X1
g17853	2.47	copper chaperone for superoxide dismutase, chloroplastic			
g23747	1.52	chaperone protein dnaJ 1, mitochondrial isoform X1			
g11534	1.43	chaperone protein dnaJ 16 isoform X1			
g14327	1.42	chaperone protein dnaJ 16-like			
**Calmodulin**					
g9795	2.10	calmodulin-like protein 8			
g12680	1.50	calmodulin-binding transcription activator 3-like isoform X1			
g7284	1.72	calmodulin-binding, receptor-like cytoplasmic kinase 2 isoform X1			
**protein NRT1/PTR FAMILY**					
g7578	1.19	protein NRT1/PTR FAMILY 8.3-like	g23257	−2.83	protein NRT1/PTR FAMILY 5.6-like
			g25949	−2.57	protein NRT1/PTR FAMILY 5.1-like
			g13192	−3.85	protein NRT1/PTR FAMILY 7.3-like isoform X1
			g21391	−3.62	protein NRT1/PTR FAMILY 6.3-like
**ABC transporter**					
g9368	5.73	ABC transporter B family member 11-like	g26516	−3.49	ABC transporter B family member 2-like
g22460	2.52	ABC transporter B family member 21-like	g20679	−3.21	ABC transporter F family member 4-like isoform X1
g9658	1.57	ABC transporter A family member 1	g15724	−4.39	ABC transporter B family member 2-like
**GTP-binding protein**					
g14812	1.17	GTP-binding protein YPTM2	g27583	−1.56	GTP-binding protein OBGC, chloroplastic isoform X1

**Table 4 ijms-22-02564-t004:** Chloroplast genes affected by UV-B stress.

Unigene ID	Log-Fold Change	Nr.annotation	Unigene ID	Log-Fold Change	Nr.annotation
g20372	4.00	Zeaxanthin, epoxidase, chloroplastic-like	g21231	−6.36	Thioredoxin M-type, chloroplastic-like
g3048	3.18	chaperone protein dnaJ 11, chloroplastic-like	g26467	−6.71	early light-induced protein 1, chloroplastic-like
g26059	3.06	protein kinase 2B, chloroplastic-like	g17433	−6.32	magnesium-chelatase subunit ChlH, chloroplastic
g17853	2.47	copper chaperone for superoxide dismutase, chloroplastic	g14073	−5.01	protein CURVATURE THYLAKOID 1B, chloroplastic
g22647	2.20	protein kinase APK1A, chloroplastic	g12540	−5.32	anthranilate synthase beta subunit 2, chloroplastic-like isoform X1
g11261	2.52	cyanidin 3-O-rutinoside 5-O-glucosyltransferase-like	g4666	−5.97	photosystem II 22 kDa protein, chloroplastic
g2107	2.02	alpha-amylase 3, chloroplastic isoform X1	g6369	−4.06	early light-induced protein 1, chloroplastic-like
g8568	2.401	dihydropyrimidine dehydrogenase [NADP (+)]-like	g15019	−4.18	photosystemII10-kDa-polypeptide, chloroplastic-like
g20708	2.00	probable L-ascorbate peroxidase 6, chloroplastic	g18469	−4.14	Phosphoglycolate-phosphatase1B, chloroplastic-like
g23545	2.69	peroxisomal (S)-2-hydroxy-acid oxidase GLO1-like, partial	g24151	−4.45	Glutaminesynthetase-leaf, isozyme, chloroplastic, partial
g27808	2.57	leucine--tRNA ligase, cytoplasmic	g13326	−3.24	photosystem I reaction center subunit V, chloroplastic-like
g24000	2.74	short-chain, type dehydrogenase/reductase-like	g2387	−3.04	phytoene synthase 2, chloroplastic-like
g4958	2.02	protein DJ-1 homolog B-like	g12907	−3.22	CBS domain-containing protein CBSX1, chloroplastic-like isoform X1
g23419	2.23	sufE-like protein 2, chloroplastic	g1234	−3.69	psbP-like protein 2, chloroplastic isoform X1
g1185	2.72	pumilio homolog 4-like	g24998	−3.86	beta-amylase 1, chloroplastic
g5427	1.42	protein kinase APK1A, chloroplastic isoform X1	g8024	−3.28	serine/threonine-protein kinase STN8, chloroplastic
g26814	1.2	glyoxylate/succinic semialdehyde reductase 2, chloroplastic-like, partial	g11142	−3.88	protein CHUP1, chloroplastic-like
g18099	1.01709	phospholipase A I isoform X1	g13505	−3.17	elongation factor G-2, chloroplastic isoform X1
g18893	1.75339	cytochrome c oxidase subunit 6b-1-like	g7192	−3.36	linoleate 13S-lipoxygenase 2-1, chloroplastic-like
g16944	1.48651	Bifunctional, aspartokinase/homoserine-dehydrogenase 1, chloroplastic-like isoform X2	g28350	−3.49	Translationfactor-GUF1-homolog, chloroplastic-like, partial
g2507	1.92168	fructose-bisphosphate aldolase 1, chloroplastic-like	g24478	−3.23	protein TIC 62, chloroplastic
g571	1.57951	stress enhanced protein 2, chloroplastic	g7433	−3.15	oxygen-evolving enhancer protein 1, chloroplastic-like
g22715	1.06424	probable 6-phosphogluconolactonase 4, chloroplastic	g19570	−3.99	tuliposide A-converting enzyme 2, chloroplastic-like
g19022	1.09656	UPF0051 protein in atpA 3′region-like	g16877	−3.50	carbonic anhydrase 2 isoform X1
g17153	1.06724	oleoyl-acyl carrier protein thioesterase, chloroplastic-like	g8464	−3.68	pheophytinase, chloroplastic-like
g3464	1.12812	kinesin-like protein KIF9	g21644	−3.45	methionine aminopeptidase 1B, chloroplastic-like
g26671	1.15177	zinc finger protein MAGPIE-like			
g3448	1.38143	protein BPS1, chloroplastic-like			
g1238	1.32983	protein gar2 [Elaeis guineensis]			

**Table 5 ijms-22-02564-t005:** RT-qPCR primers used for validation experiments.

Unigene Name	Direction	Sequence (5′->3′)	Length
LOC103699391	Forward	GTTCCAGCAGATCTCCACCT	20
	Reverse	CCGGCAATTCCACTGTTTCA	20
LOC103695855	Forward	TTTCGAAAGCAGGCAACACA	20
	Reverse	TCGACTCGGTTCATGGAGAG	20
LOC103709833	Forward	TTCAACGACATCTCCCTCGT	20
	Reverse	ATATCCATGCTGCAGTCCGA	20
LOC103711915	Forward	ATGACAGGGCTGGTGAAGAA	20
	Reverse	TGAGCAGGTCCTCAAACAGT	20
LOC103715616	Forward	AGACCTCACGAAGCCAGAAA	20
	Reverse	CTCTTCCTCCTCCTCCTCCT	20
LOC103707665	Forward	CACAGATTGTCGCACTCGTT	20
	Reverse	GCCGGAATCAGGGTCATAGA	20
LOC103715998	Forward	CCAACGAAGGCTCTCAAAGG	20
	Reverse	TTGGAAGCCTCTGGTACAGG	20
LOC103719605	Forward	GCATTCCATCCCATGACACC	20
	Reverse	CCATCTTCTCTCCCTCGCAT	20
LOC103702656	Forward	TTCGAGTTCTGTGGGCTCTT	20
	Reverse	ACGGATAGCCTACTCAACGG	20
LOC103716074	Forward	AGGCTGCGTTTGTATGTTCC	20
	Reverse	TCACCCAAACAAGGCAAAGG	20

## Data Availability

RNA sequence data that support the findings of this study have been deposited under SRA BioProject accession number PRJNA622884.

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
