# Peer review of "Novel Transcriptome Study and Detection of Metabolic Variations in UV-B-Treated Date Palm (Phoenix dactylifera cv. Khalas)"

_ijms, 2021, doi:10.3390/ijms22052564_

Round 1
Reviewer 1 Report
The manuscript entitled “Novel transcriptome study and detection of metabolic variations in UV-B-treated date palm (Phoenix dactylifera cv. Khalas)” concerns a significant problem regarding the effect of UV-B radiation on date palm as increasing amounts of UV-B radiation reaches the Earth’s surface. Authors showed altered levels of metabolites and variation of associated unigenes in transcript stages due to UV-B radiation exposure of date palm.
There are some minor mistakes like capitalized words inside the sentences (e.g. lines 58, 435, 468, 505, and 506), or the missing of “and” between upregulated downregulated at line 237. More, the lines 555-557 are italic.
In the Materials and Method section, at Plant materials, it is not specified the type of UV lamp that was used, the distance between lamp and plants, and if any filters were used.
In the Materials and Method section is described the Metabolite analysis by LC-MS/MS, but in the Results sections, I was not able to see any LC-MS/MS analysis (QTOF or QTRAP). There are only two lines in the Discussion section about such analysis. I suggest adding a paragraph in the Result section.
Reviewer 2 Report
Dear Authors,
Reviewer comments ijms-1109162
The manuscript entitled „Novel transcriptome study and detection of metabolic variations in UV-B-treated date palm (Phoenix dactylifera cv. Khalas)“ represent a useful transcriptomic study based on RNAseq approach (Illumina) with qRT-PCR validation resulting in an identification and functional annotation of differentially expressed genes (DEGs) in response UV-B stress (16 hours of UV-B light of 253.7 nm and 75 μW cm-2). The manuscript is written very comprehensively for the readers, it provides detailed results on the DEGs based on their functional annotations according to the most affected biological processes (transcription regulation, photosynthesis, oxidative phosphorylation). The data are presented in the form of comprehensive tables and figures.
I can therefore recommend the manuscript for publication in International journal of Molecular sciences.
I have only a few comments on the manuscript present version which are provided below.
1/ In addition to the individual figures and tables providing the summary of the data about DEGs ordered according to the individual functional groups, I would recommend the authors to add a figure at the end of Discussion section or as a graphical abstract providing a complex overview of all major biological processes affecetd by UV-B light stress in date palm.
Comments on methods:
2/ Materials and methods, Plant materials, line 85: The information on the source of plant material used in the study, i.e., date palm (Phoenix dactylifera) cv. Khalas used in the study has to be added, i.e., it should be specified from which institution or natural location the plants were obtained.
3/ DEGs functional annotations, line 119: In addition to the list of databases used for functional genes annotations including the NCBI, Pfam, KOG/COG and GO databases, the database versions used for the search have to be specified by date of download.
4/ In Results section, lines 181-183, the authors wrote about the four experiemtnal variants „Khalas control 1 (CR1), Khalas control 2 (CR2), Khalas UV treatment 1 (UR1), and Khalas UV treatment 2 (UR2).“ The authors should explain the reasons or differences between CR1 and CR2 as well as UR1 and UR2 variants?? Are there just biological replicates of the experimental variants or something else?
5/ Formal comments on the text:
Introduction, line 44: Replace the word „of“ by „to“ in the sentence „In full sun or partial shade, date palm trees grow and are tolerant to drought.“
Introduction, line 50: Remove the word „it“ in the sentence „UV-B radiation is a powerful environmental factor that affects many aspects of the life of a plant,…“
Introduction, line 59: Modify the word form „product“ to „produce“ in in the sentence „Under UV radiation, plants produce high levels of many amino acids and some their derivatives such as Glu, Ala, Lys, Phe and γ-amino butyrate.“ = all compounds listed are amino acids both protein amino acids as well as non-protein one (γ-amino butyrate).
Introduction, line 63: Replace the word „the development“ by „the synthesis“ in „the synthesis of functional secondary metabolites“ and remove the verb „has improved“ at the end of the sentence „This altered activity occurs after UV-B treatment at the primary metabolite stage, leading to the synthesis of functional secondary metabolites that can defend against against UV-B damage..“
Materials and methods, line 154: Remove the first word „method“ in the sentence „..and relative expression was calculated using the 2∆(-Ct) method.“
Materials and methods, line 159: Replace the heading „metabolite analysis by LC-MS/MS“ on a separate line.
Results, line 237: Add the word „and“ between the words „upregulated“ and „downregulated“ in the sentence „…the expression of 10249 and 12426 gene was upregulated and downregulated, respectively.“
Results, line 259: Add a semicolon between the words „UV light“ and „Table 2“.
Results, line 264: Remove the extra word „was“ (twice in one sentence) in the sentence „…whose expression was sixfold-lower than that in the control.“
Results, Figure 5 legend, line 303: Correct the typing error in the plant name „date palm leaves“ (not „data palm“) and in „red colour“ (not „read colour“) and modify the sentence as follows: „Detecetd metabolites are indicated with red colour.“
Results, Table 3 heading, line 325: Use a plural form of the words „transcription factors“ in the table heading, i.e., „Identified transcription factors in date palm leaf tissues under UV-B stress.“
Results, Figure 7 legend, line 364: Modify the verb „effected“ to „affected“ in the Figure 7 legend: „Photosynthesis related genes as affected by UV-B stress.“
Results, Figure 8 legend, line 376: Modify the verb „effected“ to „affected“ in the Figure 8 legend: „Oxidative phosphorylation related genes as affected by UV-B stress.“
Results, line 383: Write „compliant“ with small „c“ since it is in the sentence „..which was compliant with RNA-seq data…“
Discussion, line 415: Remove the reference „Banerjee J. et al. 2010“ since the corresponding reference number „36“ is provided at the end of the sentence.
Discussion, line 446: Remove „the“ in the senttence „…that had significantly different levels compared with those of control to explore teh relations between UV-B radiation and metabolic processes.“
Discussion, line 452: Correct the term „the unigenesis“ to „the unigenes“ in the sentence „Therefore, the changes in metabolite contents were related to the difference in the unigenes transcript levels involved in metabolite pathway responses to UV-B stress.“
Discussion, line 556: Correct the font type, i.e., remove italics from the whole senetcne starting with the words „Furthermore, the expression of most photosynthesis-related genes…“
Discussion, line 568: The sentence has to be reformulated to become clear which transcripts were upregulated and which ones were downregulated under UV-B stress „most- and least-expressed genes, transcription factor-encoding genes, chloroplast-related genes and photosystem-related genes“ - the statement has to be reformulated to become clear which transcripts were upregulated and which ones were downregulated under UV-B stress.
Final recommendation: Accept after a minor revision.
